# Synthetic Seed Production and Slow Growth Storage of In Vitro Cultured Plants of *Iris pallida* Lam.

**Annalisa Meucci** [1,*] 🆔, **Cristina Ghelardi** [1], **Giorgiana Chietera** [2] **and Anna Mensuali** [1]

1   Crop Science Research Center, Scuola Superiore Sant'Anna, Piazza Martiri della Libertà 33, 56127 Pisa, Italy; cristina.ghelardi@santannapisa.it (C.G.); anna.mensuali@santannapisa.it (A.M.)
2   LMR Naturals by IFF (International Flavors & Fragrances SAS), Parc Industriel des bois de Grasse, 18/20 Avenue Joseph Honoré Isnard, 06130 Grasse, France; giorgiana.chietera@iff.com
*   Correspondence: annalisa.meucci@santannapisa.it

**Abstract:** *Iris pallida* Lam. is traditionally cultivated in Italy to sell its rhizomes to perfume-producing industries and is particularly sought-after because of its high content of irones, ketone compounds responsible for the violet smell of the orris essence. One of the critical aspects of its cultivation is the propagation method, performed by subdividing and replanting sections of the rhizome, which leads to the sacrifice of salable material. A solution is provided via in vitro propagation using the somatic embryogenesis technique, an effective method that allows the production of plants without the use of the rhizome. To facilitate the scale up of the activities of micropropagation companies, the method of slow growth storage (SGS) for orris plantlets and a somatic embryo encapsulation technique were developed for the first time. Orris plantlets were placed at 4 °C in the dark for 30, 60, 90 and 120 days and monitored 7 and 30 days after treatment. Synthetic seeds were obtained by encapsulating somatic orris embryos in sodium alginate beads, which were stored for 14 and 28 days at 4 °C and 24 °C. The results showed that it is possible to cold-preserve orris plantlets for up to 90 days without significant damages and that orris synthetic seeds can be produced and stored for a short-to-mid-term period. These conservation techniques can be useful for germplasm conservation and can also be integrated in the micropropagation cycle of orris, helping to solve issues related to the traditional propagation method.

**Keywords:** orris; micropropagation; somatic embryogenesis; cold conservation

## 1. Introduction

*Iris pallida* Lam. is one of the most representative species of the Mediterranean area and, thanks to its characteristic flowers, contributes in improving the landscape value of Italian hills and mountains. This species has been cultivated in Italy for centuries and now is mainly cultivated in France, Morocco, and China for perfume production [1], although it is also widely used in the pharmaceutical field [2] and in the production of spirits. *I. pallida* is one of the most frequently used plant species in the perfume field as its essence is one of the most commonly used in perfume formulations; this is the reason why this plant is particularly sought after on both national and European markets. Orris essence is obtained via the distillation of its rhizome, the part of the plant with the greatest commercial value, which is sold to industries traditionally involved in the production of essential oils and perfumes, mainly based in France. For 3–4 years of cultivation in the field, triterpenoid compounds called iridals (iripallidal and iriflorental) accumulate inside rhizome tissues [3,4]. Then, rhizomes are collected and stored in ventilated rooms for another 3–4 years; during this period, iridals oxidize and become irones, ketone compounds responsible for the typical violet scent of the essence. At the end of postharvest conservation, the rhizome undergoes two rounds of hydrodistillation, which allow the obtention of the orris butter, with an average irone content of 20%, and then the obtention of the precious

orris essence, which can reach an irones content of 80% [1]. Since the orris essence is important for the luxury market, cultivating *I. pallida* represents an important source of income for many rural areas all over the world, despite the fact that its production chain presents some critical aspects that discourage its cultivation. One of the main difficulties concerns the propagation method traditionally used, which requires the use of part of the rhizome and involves the sacrifice of a high amount of raw material that could be potentially sold to the perfume industry. The ability to obtain new orris plantlets using the in vitro micropropagation method would be an effective solution to this issue. Specifically, the somatic embryogenesis technique performed on explants of immature flower buds has already given promising results [5,6]. However, in vitro conservation protocols for the propagation material of this species have not been investigated so far. The development of these techniques would facilitate production planning in nursery companies and allow both the maintenance of in vitro banks and the exchange of germplasm orris resources. The in vitro conservation of plant material can be performed using several techniques that mostly differ according to the type of explant used (embryos, nodes, apices, or shoots) and the environmental conditions adopted (cold storage or cryoconservation). The slow growth storage (SGS) methodology was already tested in several micropropagated species [7–10]. Using synthetic seed technology through the alginate encapsulation of somatic embryos is an effective procedure for short-to-mid-term storage and the exchange of germplasms [11–13].

Based on these premises, this study aims to achieve the in vitro medium-term conservation of *I. pallida* using synthetic seeds and applying slow growth storage (SGS) for micropropagated plantlets.

## 2. Materials and Methods

### 2.1. Induction of Somatic Embryos and Plantlet Regeneration

Micropropagation of *I. pallida* was performed using the in vitro somatic embryogenesis technique, in accordance with the protocol of Lucchesini et al. [6]. Immature flower buds taken from *I. pallida* cv. RQ plants provided by the International Flavors and Fragrances Industry (LMR-IFF) based in Grasse (France) were deprived of their outer sheath and rinsed with tap water for 20 min; then, they were subjected to a disinfection protocol carried out using 70% EtOH and 15% NaClO. Explants consisting of the central petals, and the youngest and least contaminated parts of the bud were obtained under a laminar flow hood and cultured on a callus induction medium composed of a Murashige & Skoog medium [14], supplemented with 500 mg $L^{-1}$ of 2-(N-morpholino) ethane sulfonic acid (MES), 300 mg $L^{-1}$ of reduced glutathione (GSH), 3.0 g $L^{-1}$ of gelrite® (Merck Life Science S.r.l., Milan, Italy) (basal medium), 50 g $L^{-1}$ of sucrose, 1 mg $L^{-1}$ of 2,4-dichlorophenoxyacetic acid (2,4 D), and 1 mg $L^{-1}$ of Kinetin (i1A medium). The yellow calluses that formed were then transferred to an embryo induction medium composed of the basal medium, 50 g/L of sucrose, 1 mg $L^{-1}$ of 2,4 D, and 0.1 mg $L^{-1}$ of Kinetin (i1B medium). Therefore, embryos were moved onto a specific germination medium composed of the basal medium with Gamborg B5 vitamins [15], 20 g $L^{-1}$ of sucrose, 0.1 mg $L^{-1}$ of 6-benzyladenine (BA), and 0.1 mg $L^{-1}$ of α-naphthaleneacetic acid (NAA) (i2 medium). Finally, micropropagated plantlets were grown in containers (four plants each) with 50 mL of i2 medium containing 30 g $L^{-1}$ of sucrose (i3 medium), and subcultured every three weeks. The pH of every medium was adjusted to 5.9 with 1N KOH, and media were all autoclaved at 121 °C (ca. 0.12 MPa) for 15 min. Plantlets were maintained in a growth chamber at 22 °C, under a 16 h photoperiod (100 µmol $m^{-2}s^{-1}$). During the last phase of micropropagation, cycle orris plantlets rooted and further acclimatized.

### 2.2. Slow Growth Storage (SGS)

To test cold tolerance, at the end of each subculture period, micropropagated plantlets (8 for each treatment) were stripped of roots and old leaves and placed on i3 medium at 4 °C in the dark for 30, 60, 90, and 120 days. At the end of each period (0 days), plantlets were moved from the laboratory refrigerator to the growth chamber for 7 days before being

subcultured on the same medium for 23 days. Morphological and biochemical parameters were determined right after plantlet removal from cold storage (0 days), as well as 7 and 30 days after the end of SGS. Plantlets at the end of each standard subculture period not subjected to cold storage were used as a control (Ctr).

Morphological and Biochemical Determinations

The number of roots and new leaves were monitored, and the quantitative determination of total chlorophyll and carotenoid contents was performed spectrophotometrically on both treated and Ctr plantlets. Briefly, 20 mg of leaf tissue for each replicate (7 replicates for each treatment) were extracted and placed in 1,5 mL of methanol (100% ($v/v$)) for 24 h at 4 °C in the dark. Absorbance was read at $\lambda$ = 665.2 nm and Chl b, $\lambda$ = 652.4 nm for chlorophylls a and b, respectively, and at $\lambda$ = 470 nm for carotenoids. Pigment levels were calculated using Lichtenthaler's formula and expressed as the fresh weight of the tissue [16].

### 2.3. Synthetic Seeds Production and Storage

Somatic embryos (averaging approximately 2 mm in length) developed in the i2 medium were used for the preparation of synthetic seeds. For encapsulation, embryos were mixed within a $Ca^{2+}$-free MS medium supplemented with 30 g $L^{-1}$ of sucrose, 300 mg $L^{-1}$ of GSH, 500 mg $L^{-1}$ of MES, and 30 g $L^{-1}$ of sodium alginate (3%), and then dropped using a pipette into a complexing solution made of MS 0 and 14.7 g $L^{-1}$ of $CaCl_2 \cdot 2H_2O$ (100 mM). The pH of all media and solutions used in the experiment was adjusted to 5,6, and all the solutions were autoclaved at 121 °C for 20 min. For the polymerization of sodium alginate, the beads were held in the complexing solution for about 30 min under continuous stirring. After polymerization, the hardened alginate beads were gently rinsed twice, for 15 min each time, with sterile water, and then placed in Petri dishes (Ø 50 mm) with sterile filter paper inside to eliminate excess water; the Petri dishes were then hermetically sealed with parafilm to avoid drying. During the storage period, the encapsulated *I. pallida* embryos were maintained in Petri dishes (Ø 50 mm, 6 capsules dish$^{-1}$) with filter paper soaked with sucrose-free half-strength MS minerals supplemented with 300 mg $L^{-1}$ of GSH and 500 mg $L^{-1}$ of MES. Five Petri dishes per treatment (5–6 synthetic seeds per dish) were maintained for each storage period. The Petri dishes were stored in darkness under the following conditions:

(A)   Fourteen days of storage at a cold temperature of 4 °C in a laboratory refrigerator;
(B)   Fourteen days of storage at a room temperature of 23 °C;
(C)   Twenty-eight days of storage at a cold temperature of 4 °C in a laboratory refrigerator;
(D)   Twenty-eight days of storage at a room temperature of 23 °C;

At the end of the storage period, synthetic seeds were sown and maintained as the control (0 days of storage). The control consisted of *I. pallida* embryos sowed immediately after encapsulation on i2 germination medium. Four weeks after sowing on the embryo germination medium, the following parameters were measured: number of browned beads, number of embryos that develop into seedlings (germination of the synthetic seeds), and number of new shoots per bead.

### 2.4. Statistical Analysis

All data were expressed as the mean $\pm$ S.E. of the number of replicates. A Student *t*-test or analysis of variance (ANOVA) was performed at a $p \leq 0.05$ significant level. The mean values were separated using the Tukey post-test ($p < 0.05$). Statistical analysis was carried out with GraphPad Prism (version 10.00 for Windows, GraphPad Software, La Jolla, San Diego, CA, USA).

## 3. Results

### 3.1. Slow Growth Storage (SGS)

Once removed from the refrigerator, micropropagated plants subjected to 30 days of SGS did not show significant visual damages; those treated for 60 and 90 days showed moderate yellowing, and plantlets treated for 120 days showed severe yellowing (Figure 1); plantlets subjected to 60 and 90 days of coldness were able to produce new roots during the cold treatment (Figure 2). All plantlets, except those subjected to 120 days of SGS, resumed root production as soon as 7 days after the end of the treatment. Thirty days after the treatment, plantlets belonging to T30, T60, and T90 treatments produced a significantly greater number of roots compared with that produced seven days after the treatment, reaching the average number of roots of the control plantlets.

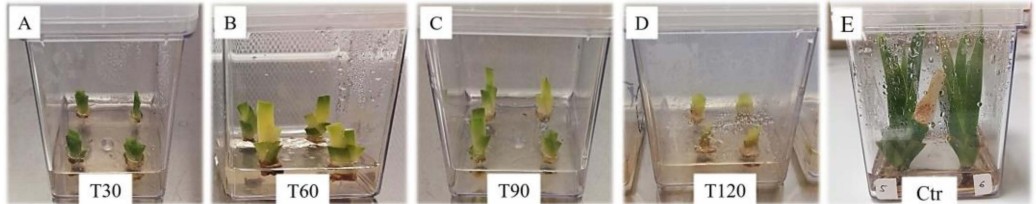

**Figure 1.** Plantlets appearance after 30 (**A**), 60 (**B**), 90 (**C**), and 120 (**D**) days of SGS (T30, T60, T90, and T120) and not stored control (Ctr) plantlets (**E**).

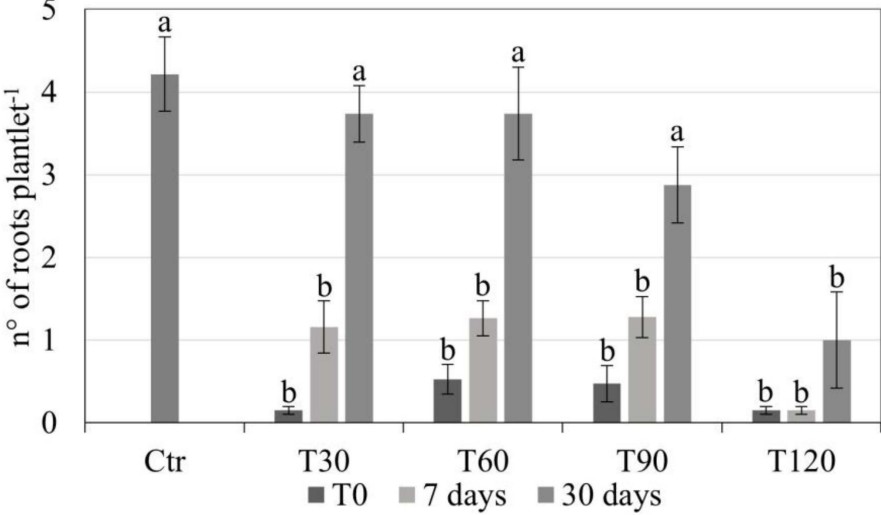

**Figure 2.** The average number of roots of *I. pallida* cv. RQ plantlets right after their removal from cold storage (T0), 7 and 30 days after 30, 60, 90 and 120 days of SGS (T30, T60, T90, and T120), compared with the number of roots of plantlets not stored in the cold produced in a single subculture (Ctr). Data, reported as mean values ± S.E., were subjected to an analysis of variance (ANOVA), and the different letters indicate significant differences between the treatment and Ctr, within each sampling date (7 and 30 days) (Tukey's post-test, $p \leq 0.05$).

After being removed from cold storage, plantlets belonging to the T60 and T90 treatments already showed leaf elongation compared with those treated for 30 days (Figure 1A–C). Concerning the number of new leaves produced, cold-treated plantlets, regardless of the treatment duration, started to produce them 7 days after the end of each experiment; moreover, T60 plantlets had already reached the average number of new leaves of control plantlets by this point (Figure 3). Thirty days after each treatment, all other cold-treated plantlets showed a significant increase in leaf development, reaching values comparable to those of control plantlets (Figure 3).

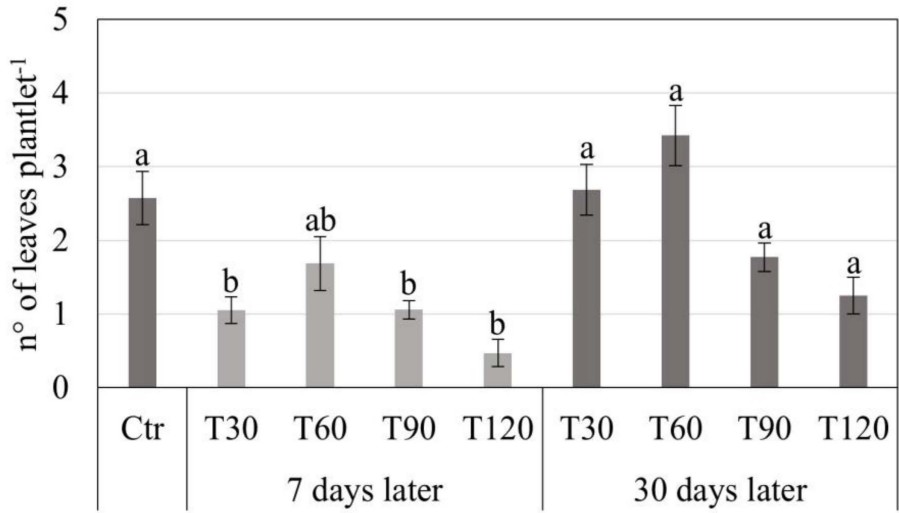

**Figure 3.** The average number of new leaves of *I. pallida* cv. RQ plantlets right after their removal from cold storage (T0), 7, and 30 days after 30, 60, 90 and 120 days of SGS (T30, T60, T90, and T120), compared to that of new leaves of plantlets not stored in the cold grown in a single subculture (Ctr). The data, reported as mean values ± S.E., were subjected to an analysis of variance (ANOVA), and the different letters indicate significant differences between the treatment and Ctr, within each sampling date (7 and 30 days) (Tukey's post-test, $p \leq 0.05$).

The analyses of chlorophyll and carotenoid content (Figures 4 and 5) demonstrate that 7 days after SGS, regardless of the treatment duration, plantlets showed a reduction in both chlorophyll and carotenoid content, corresponding to the days of SGS. Thirty days after SGS, plantlets stored for 30, 60, and 90 days showed an increase in both chlorophyll and carotenoid content, reaching values comparable to those of the control plantlets. However, the prolonged exposure of plantlets to cold conditions for 120 days resulted in an irreversible impairment of chlorophyll and carotenoid contents.

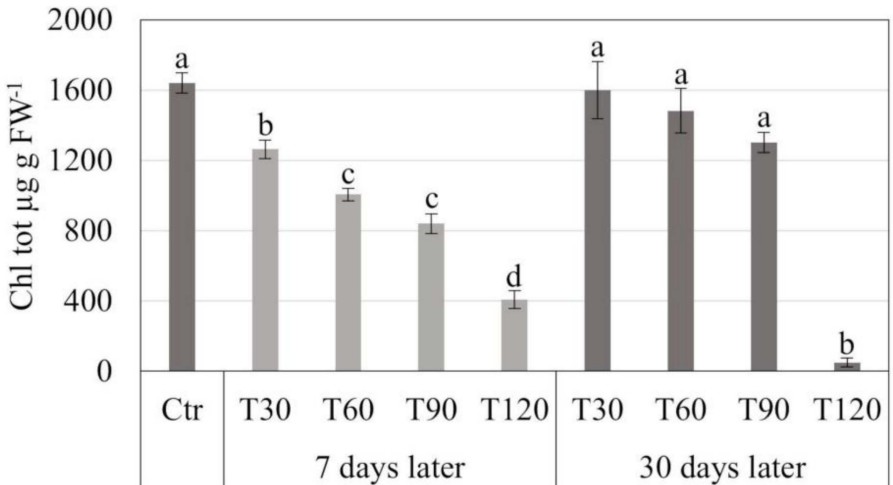

**Figure 4.** Total chlorophyll (a + b) content (μg g Fresh Weight$^{-1}$) in *I. pallida* cv. RQ plantlets, 7 and 30 days after their removal from SGS (T30, T60, T90, and T120), compared with that of non-refrigerated plantlets (Ctr). The data, reported as mean values ± S.E., were subjected to an analysis of variance (ANOVA), and the different letters indicate significant differences between treatments, within each sampling date (7 and 30 days later), and the Ctr. (Tukey's post-test, $p \leq 0.05$).

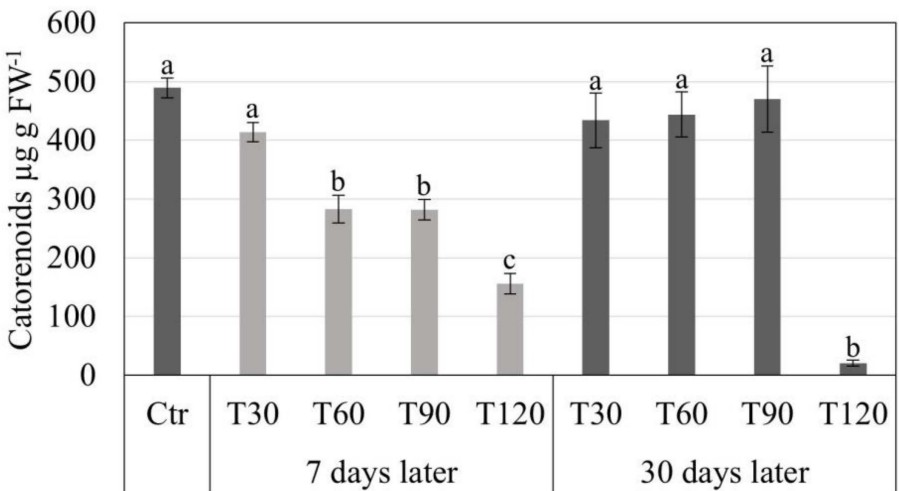

**Figure 5.** Carotenoid content ($\mu g$ g Fresh Weight$^{-1}$) in *I. pallida* cv. RQ plantlets, 7 and 30 days after their removal from SGS (T30, T60, T90, and T120), compared with that of plantlets not stored in the cold (Ctr). The data, reported as mean values $\pm$ S.E., were subjected to an analysis of variance (ANOVA), and the different letters indicate significant differences between treatments, within each sampling date (7 and 30 days later), and the Ctr (Tukey's post-test, $p \leq 0.05$).

### 3.2. Synthetic Seed Production and Storage

The 3% alginate beads were hard, transparent, and symmetrical (Figure 6B). When the synthetic seeds were immediately cultivated on the embryo germination medium for 4 weeks, they revealed that 30% of embryos were browned and that nearly half of them were able to develop into seedlings, producing an average of 1.5 new shoots from each bead (Figure 6C–E). If the synthetic seeds were stored, it was found that the browning percentage of the encapsulated embryos after 4 weeks on the germination medium (Figure 7) was 50% if they were stored for 28 days, but when seeds were conserved for 14 days, the damage was limited to less than 10%, particularly when seeds were kept at 4 °C.

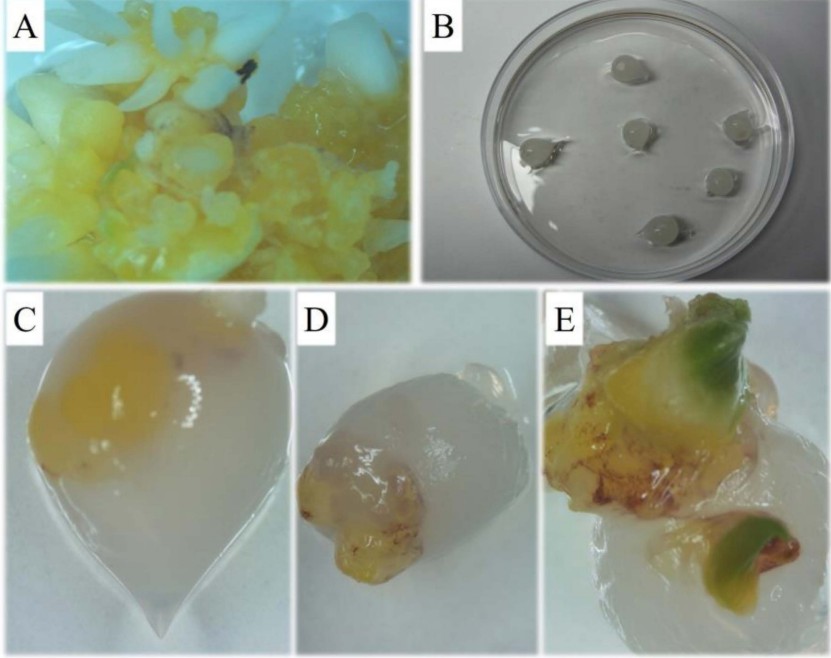

**Figure 6.** Synthetic seed production from somatic embryos of *I. pallida*. (**A**) Somatic embryos developed on embryogenic callus; (**B**) gelled beads on medium solidified with agar; (**C–E**) germination process from synthetic seeds on agar-solidified medium.

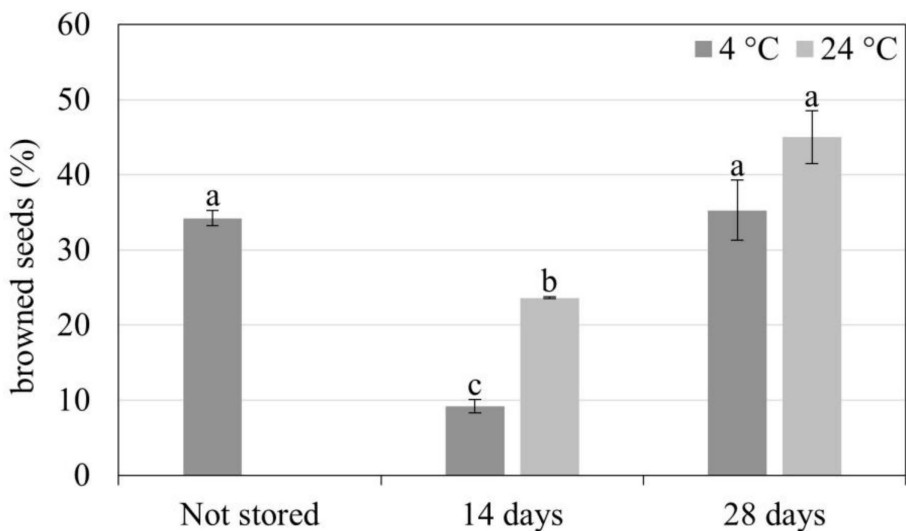

**Figure 7.** Percentage of browning on *I. pallida* synthetic seeds, 14 and 28 days after storage at 4 °C or 24 °C, compared to that on control ones (not stored). The data, reported as mean values ± S.E., were subjected to an analysis of variance (ANOVA), and the different letters indicate significant differences among means (Tukey's post-test, $p \leq 0.05$).

Encapsulated embryos maintained a good germination ability if stored at 4 °C, while room temperature reduced their germinability to 40% (Figure 8). Following embryo germination, conservation dramatically reduced the number of new shoots produced per embryo as compared with the number of shoots developed by an encapsulated embryo immediately grown on the germination medium (Figure 9). At 4 °C, the number of shoots per embryo considerably decreased on the 14th day.

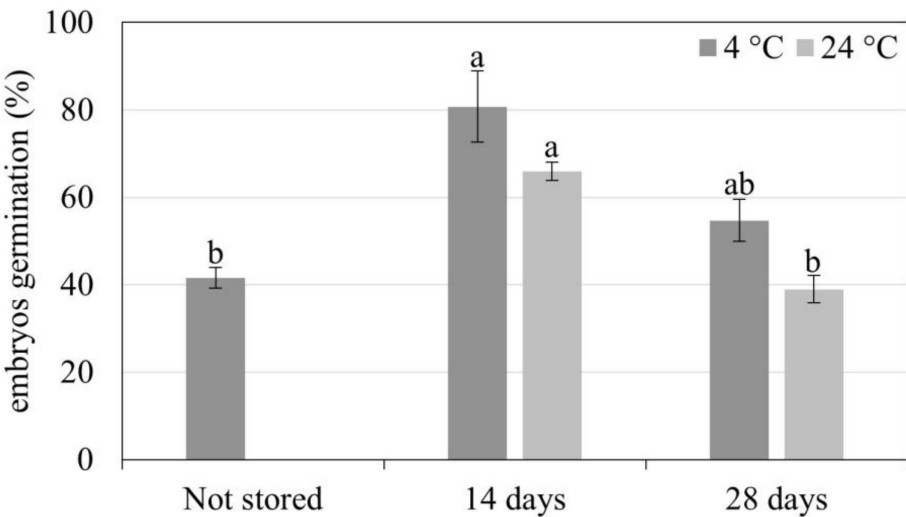

**Figure 8.** Germination ability of *I. pallida* synthetic seeds, 14 and 28 days after storage at 4 °C or 24 °C, compared with that of control ones (not stored). The data, reported as mean values ± S.E., were subjected to an analysis of variance (ANOVA), and the different letters indicate significant differences among means (Tukey's post-test, $p \leq 0.05$).

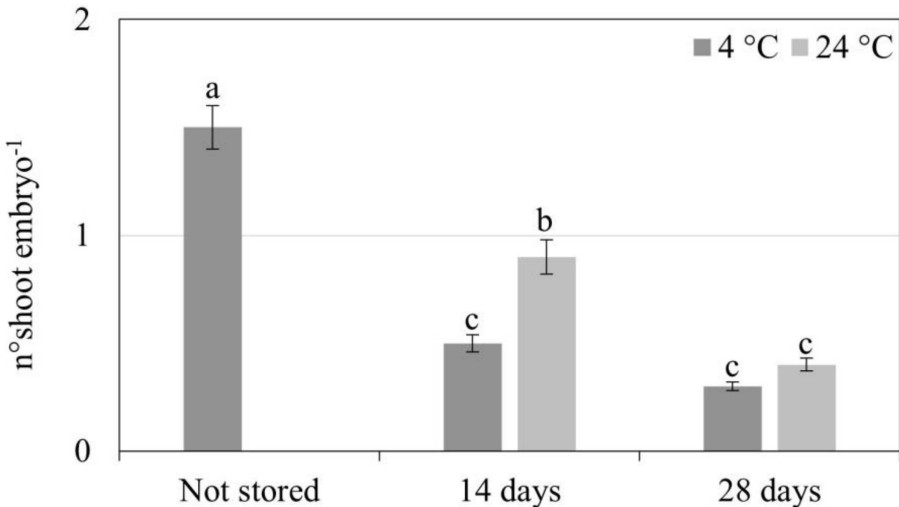

**Figure 9.** Number of developing shoots in *I. pallida* synthetic seeds, 14 and 28 days after storage at 4 °C or 24 °C, compared with that in control ones (not stored). The data, reported as mean values ± S.E., were subjected to an analysis of variance (ANOVA), and the different letters indicate significant differences among means (Tukey's post-test, $p \leq 0.05$).

## 4. Discussion

The in vitro slow growth storage (SGS) of shoot cultures represents an useful tool for the conservation of plant genetic resources and for improving the micropropagation of ornamental species by reducing [17] the plant growth rate and, consequently, increasing the subculture periods [18]. The application of this technique can also allow for better production planning and the better management of nursery production.

In particular, when a greater number of produced plants is required in specific periods of the year, the storage of micropropagated plants in a cold chamber, before their acclimatization, allows for an improvement in the organization of company activities, as is the case for *I. pallida*, which requires 200,000 plants ha$^{-1}$ on average for the initial planting [19]. Moreover, stored plants can constitute a reserve of healthy and uncontaminated material in the case of unexpected contaminations during the in vitro production process or in the climatic chambers. The SGS technique is based on the fact that plant metabolism reduces its functions at low temperatures; the temperature used for this conservation must be able to reduce metabolic processes without compromising the ability of plants to restore their normal condition [20]. The temperatures used for non-tropical species vary between 2 °C and 12 °C, while 4–5 °C is the most commonly used range for in vitro conservation [21]. Another important factor is the absence of light; cold-stored plants are kept in conditions of low light intensity [22] or in total darkness, which we applied during the experiment on *I. pallida*. The darkness condition interacts with the low temperature to maintain a slow growth rate, and, at the same time, this can represent a reduction in energy costs. In the literature, many studies have been developed on the composition of the culture medium to reduce plant growth under cold conditions [22,23]. During the storage period, *I. pallida* plantlets were cultured on a medium (i5) that contained both osmoregulators (30 g L$^{-1}$ sucrose) and antioxidants (reduced glutathione, GSH). This allows for the maintenance of the stability and quality of plantlets, as they are not affected by the stress related to nutrient depletion and protected from browning phenomena. For this purpose, in other studies, additional treatments such as ABA or specific osmotic agents were used during cold storage [24]. The SGS treatment was proven to be an effective method for preserving micropropagated orris plantlets for up to 90 days; in fact, plantlets were able to re-adapt to the environmental conditions in just 30 days and resumed the regular growth of the apical portion. In the literature, longer periods of slow conservation are reported; for example, a study conducted on *Anthurium* sp. and *Ranunculus* sp. demonstrates how cold storage can be useful in sprout preservation, for up to 8 and 9 months, respectively [18].

It is therefore also possible to prolong orris conservation by using specific treatments, even if 90 days can be considered a proper period of conservation, to avoid the risk of somaclonal variations, which are observed in some cases of prolonged conservation [23]. The recovery of plantlet vitality was further confirmed via the contents of total chlorophylls and carotenoids, which 30 days after the treatment, were similar to those of the control group. The re-establishment of both chlorophylls and carotenoids in treated plantlets can positively affect their acclimatization [25]. The favorable regrowth of orris plantlets was likely facilitated by effective gas exchange, attributed to the use of Magenta® (Merck Life Science S.r.l., Milan, Italy) vessels with lids, enabling higher gas exchange compared with that when using a traditional gastight in vitro culture container [26,27]. The utilization of these vessels prevents ethylene accumulation and excessive humidity, factors associated with physiological and morphological disorders such as hyperidricity, which can adversely affect plantlets' ability to withstand acclimatization stress [28].

The use of synthetic seeds represents a step forward in conservation techniques, as it allows for the easy handling, shipping, and ex situ conservation of elite plants [29]. Recently, axillary buds, shoot tips, and nodes have been utilized in synthetic seed production [8], although somatic embryos are predominantly employed in this technique [30,31]. The explants undergo immersion in a sodium alginate solution and are subsequently placed in a calcium chloride solution to facilitate the formation of the alginate capsule. The balance between these two solutions is fundamental, and the ion exchange between them varies depending on the explant and the species to be encapsulated [32]. At concentrations of less than 3% for sodium alginate and less than of 100 mM for $CaCl_2$, the alginate beads in *Teucrium polium* L. were very loose and unable to cover the plant tissue, while concentrations higher than 3% resulted in the darkening of plant tissues and the absence of germinated plants [33]. Concentrations of 3% for sodium alginate and 100 mM for $CaCl_2$ were chosen for the production of synthetic seeds of *I. pallida* species, effectively facilitating the formation of alginate beads coating the plant tissues without negatively impacting the regrowth of somatic embryos. The alginate solution serves as an artificial endosperm [34]. MS medium was employed to create the alginate beads around orris embryos, allowing for the absorption of medium salts, vitamins, and sugar to encourage embryo development into new plantlets. Ascorbic acid, along with other antioxidants, can be added as an additive in synthetic seed cultures [9]. The culture medium used for both synthetic beads and the regrowth of *I. pallida* plantlets was supplemented with GSH, preventing and containing browning processes, which characterized the entire procedure, as observed in the results. Concerning synthetic seed conservation, the literature highlights two key factors: the temperature adopted and the storage period [35,36]. In most published papers, the incubation temperature used was higher than 0 °C, proving this to be effective not only for tropical species, which are better stored at room temperature [37], but also for the majority of studied species [38]. Cold storage at 4 °C for 14 days has been shown to improve germination in many horticultural species [39,40], reducing metabolic activity and preserving nutritive resources to support post-storage germination. The adoption of these storage conditions facilitates the conversion of orris synthetic seeds into new plantlets. However, irrespective of the temperature and storage interval, conservation reduces the number of shoots produced by embryos 30 days after cold storage, although a slight reduction in the number of newly developed shoots can be easily overcome in the subsequent phases of mass micropropagation via shoot multiplication.

## 5. Conclusions

In conclusion, both the SGS and synthetic seed techniques demonstrated their utility in addressing critical issues associated with the traditional propagation of *Iris pallida* Lam. These methods allow for the simplification of scaling up activities related to orris in vitro production, as illustrated in Figure 10. Specifically, the conservation of orris embryos through encapsulation in synthetic beads (Figure 10D) could enhance the in vitro regeneration phase (Figure 10B–E), facilitating embryo culture and creating a potential

reserve of regenerated material. Simultaneously, the ability to cold-store micropropagated plantlets (Figure 10F) before the acclimatization process (Figure 10G) could provide the necessary number of plantlets required to meet seasonal demands from orris-production farms. Solving the critical aspects of propagation in the orris production chain would serve as a valuable incentive for the widespread cultivation of orris.

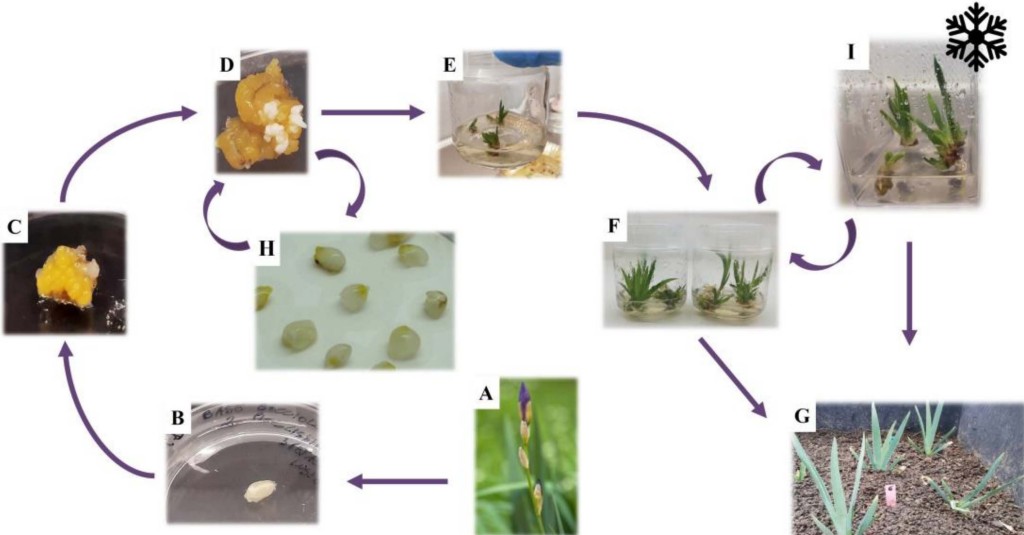

**Figure 10.** Phases of *Iris pallida* micropropagation ((**A**) flower buds in field; (**B**) immature flower buds on i1A medium; (**C**) embryogenic callus on i1A medium; (**D**) Somatic embryos formed on callus on i1B medium; (**E**) first plantlets obtained on i2 medium; (**F**) plantlet multiplication on i3 medium; (**G**) acclimatization) and possible integration of conservation techniques ((**H**) synthetic seed production with somatic embryos; (**I**) SGS of micropropagated plantlets).

**Author Contributions:** Conceptualization and methodology, A.M. (Anna Mensuali) and A.M. (Annalisa Meucci); formal analysis and investigation, A.M. (Annalisa Meucci) and C.G.; data curation, A.M. (Annalisa Meucci); writing—review and editing, G.C., A.M. (Anna Mensuali) and A.M. (Annalisa Meucci); supervision, A.M. (Anna Mensuali); funding acquisition, A.M. (Anna Mensuali) and G.C. All authors have read and agreed to the published version of the manuscript.

**Funding:** This study was carried out within the Agritech National Research Center and received funding from European Union Next-GenerationEU (PNRR) AGRITECH, PNRR M4C2, id CN00000022 (CUP J53C22001610007). This manuscript reflects only the authors' views and opinions, and neither the European Union nor the European Commission can be considered responsible for them.

**Data Availability Statement:** The raw data supporting the conclusions of this article will be made available by the authors on request.

**Acknowledgments:** Thanks go to LMR-International Flavors and Fragrances Industry (LMR-IFF) based in Grasse (France) for co-financing the research activities and providing part of the plant material. The authors acknowledge, for his contribution to some experiments related to this work, student Francesco Elia Florio, and dedicate the publication of this research to their dear colleague Mariella Lucchesini.

**Conflicts of Interest:** Author Giorgiana Chietera was employed by the company LMR Naturals by IFF (International Flavors & Fragrances SAS) and declares no conflict of interest. The remaining authors declare that the research was conducted in the absence of any commercial or financial relationships.

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
