# Peer review of "Synthetic Seed Production and Slow Growth Storage of In Vitro Cultured Plants of Iris pallida Lam."

_horticulturae, doi:10.3390/horticulturae10030272_

Round 1

Reviewer 1 Report

Comments and Suggestions for Authors

The study touches on an interesting topic for propagation and conservation of valuable plants, which can help increase the efficiency of cultivation of I. pallida and production of the orris essence. The work is of good quality and clearly presented, I just have a few comments, mainly 'cosmetic'. 

- The order of the name and surname of the second author is inverted with respect to the other authors

- Throughout the manuscript, the post-hoc test is ‘Tukey’ and not ‘Tuckey’

- Figure 1. Would it be possible to show a photo of the controls?

- In Figures 4 and 5, I understand that each set of data taken 7 or 30 days after ending conservation at 4ºC is compared to the control, and for this reason there are different data sets with the same letter (specifically 'b'), albeit not belonging to the same significance group (they are evidently quite dissimilar between themselves). This is a bit confusing at a first glance, because seeing the same letter leads one to think they belong to the same group. Could a different letter be used (for example to indicate the T120 30 days later samples) to make the graph clearer?

Line 292: ‘Anthurium’ and ‘Ranunculus’ should be in italics.

Since the first and last authors share the same initials, it would be useful to differentiate them in the Author Contributions section.

Comments on the Quality of English Language

There are only a few inaccuracies throughout the text, below I am reporting those I have spotted.

- Materials and Methods 2.1 ‘Obtention of embryos and plantlets’

- Line 79: ‘the obtained yellow calluses’

- Line 90: ‘develop the root apparatus’

- Line 101: ‘leaf explants’

- Line 143: the plural of ‘thesis’ is ‘theses’

Line 309: remove ‘to’ (‘explants undergo an immersion’)

- Line 347: ‘before the acclimatization process’

Reviewer 2 Report

Comments and Suggestions for Authors

The authors presented the article describing the slow growth storage as well as the production of synthetic seeds of Iris pallida, plant species traditionally cultivated in Italy. It is very important to note that a slow growth storage of iris plantlets and a somatic embryos encapsulation technique have been developed successfully for the first time. The interesting results presented in this work showed that it is possible to cold-preserve iris plantlets up to 90 days without significant damages. Also, iris synthetic seeds can be produced and stored for short mid-term.

Introduction part is correctly written with sufficient literature data. The experimental methods used in this work are described but without all sufficient details (all recommendations are incorporated directly in the text manuscript). The obtained results are discussed sufficient and are followed by a very nicely illustrated conclusion.

All of my detailed corrections are incorporated directly in the text manuscript.

Reviewer 3 Report

Comments and Suggestions for Authors

Subject: Reviewer's Feedback for Manuscript titled "Synthetic Seeds Production and Slow Growth Storage of In Vitro Cultured Plants of Iris pallida Lam."

Dear Authors,

I have had the opportunity to carefully review your manuscript titled "Synthetic Seeds Production and Slow Growth Storage of In Vitro Cultured Plants of Iris pallida Lam." I appreciate the effort and dedication put into the study. The manuscript provides valuable insights into synthetic seed production and slow growth storage techniques for in vitro cultured plants of Iris pallida Lam. While the applied techniques and procedures in in vitro conservation may not introduce novel methodologies, the obtained results are significant. The protocols presented in your study have the potential to play a role in germplasm conservation. Moreover, the integration of these protocols into the micropropagation cycle of orris is noteworthy, addressing challenges associated with traditional propagation methods. The practical applications of your findings are commendable, as they contribute to the advancement of conservation practices and offer solutions to issues faced in orris propagation.

Given the structure of the document and the volume of comments, it was challenging to provide an exhaustive review directly in the manuscript. Consequently, I have compiled all my comments in a PDF document attached herewith.

It is crucial that the authors thoroughly review the attached document and carefully address all the comments. Only after incorporating the suggested changes, I can confidently recommend the manuscript for publication. Please be aware that this meticulous consideration is essential to ensure the clarity and accuracy of the results and interpretations.

Thank you for your commitment to this research, and I look forward to the revised version of your manuscript.

Comments on the Quality of English Language

A significant portion of the feedback pertains to language usage, specifically in terms of clarity and precision. I made efforts to enhance the English language presentation to avoid any potential confusion in the presentation and discussion of results. I strongly recommend that the authors consider sending the manuscript for professional English editing to further refine the language.
